# Size-dependent ice nucleation by airborne particles during dust events in the Eastern Mediterranean

Naama Reicher[1], Carsten Budke[2], Lukas Eickhoff[2], Shira Raveh-Rubin[1], Ifat Kaplan-Ashiri[3], Thomas Koop[2] and Yinon Rudich[1]*

[1] Department of Earth and Planetary Sciences, Weizmann Institute of Science, 76100 Rehovot, Israel
[2] Faculty of Chemistry, Bielefeld University, Universitätsstraße 25, 33615 Bielefeld, Germany
[3] Chemical Research Support, The Weizmann Institute of Science, 76100 Rehovot, Israel

*Correspondence to*: Yinon Rudich (yinon.rudich@weizmann.ac.il)

**Abstract.** Prediction of cloud ice formation in climate models remains a challenge, partly due to the complexity of ice-related processes. Mineral dust is a prominent aerosol in the troposphere and is an important contributor to ice nucleation in mixed phase clouds, as dust can initiate ice heterogeneously at relatively low supercooling conditions. We characterized the ice nucleation properties of size-segregated mineral dust sampled during dust events in the Eastern Mediterranean. The sampling site allowed to compare between the properties of airborne dust from several sources with diverse mineralogy that passed over different atmospheric paths. We focused on particles with six size-classes, determined by the Micro-Orifice Uniform Deposit Impactor (MOUDI) cut-off sizes: 5.6, 3.2, 1.8, 1.0, 0.6 and 0.3 μm. Ice nucleation experiments were conducted in the WeIzmann Supercooled Droplets Observation on Microarray (WISDOM) setup, where the particles are immersed in nanoliter droplets using a microfluidics technique. We observed that the activity of airborne particles depended on their size-class, where supermicron and submicron particles had different activities, possibly due to different composition. The concentrations of ice nucleating particles and the density of active sites ($n_s$) increased with the particle size and particles concentration. The supermicron particles in different dust events showed similar activity, which may indicate that freezing was dominated by common mineralogical components. Combining recent data of airborne mineral dust, we show that current predictions, which are based on natural dust or standard mineral dust, overestimate the activity of airborne dust, especially for the submicron class, and therefore we suggest to include information of particle size in order to increase the accuracy of ice formation modelling and, thus, in weather and climate predictions.

## 1 Introduction

Cloud droplets can supercool to 238 K before homogeneous freezing occurs (Koop and Murray, 2016;Rosenfeld and Woodley, 2000). At warmer temperatures, heterogeneous ice nucleation (HIN), where the presence of aerosol particles lowers the required energy barrier to form a stable ice nucleus is the common pathway of ice formation (Murray et al., 2012;Pruppacher

and Klett, 1997;Khvorostyanov and Curry, 2004;Hoose and Möhler, 2012). These ice-nucleating particles (INPs) can be activated at sub-zero temperatures and subsequently lower humidity conditions, mainly by interaction with supercooled droplets. INPs are relatively rare particles and comprise only about $10^{-5}$ of the total ambient particles in the free troposphere (Rogers et al., 1998). Yet, their interaction with clouds can greatly influence climate (Gettelman et al., 2012;Tan et al., 2016;Lohmann and Feichter, 2005). Therefore it is important to represent them well in weather and climate models (DeMott et al., 2010). Currently, ice formation is a source of great uncertainty in cloud and climate models, partly due to the complexity of ice processes and the insufficient understanding of the key surface properties which determine an INP (IPCC, 2013). To improve the predictions of models, a great effort is invested in the characterization of INPs and in the development of parametrizations based on their physical and chemical properties (Cantrell and Heymsfield, 2005;Niemand et al., 2012;Ullrich et al., 2017).

One of the most abundant INP in the atmosphere is mineral dust, which originates in dryland zones, such as deserts (Middleton, 2017;DeMott et al., 2003b). Field observations have identified an increase in INP concentrations and ice clouds formation in the presence of mineral dust (Ansmann et al., 2008;Rosenfeld et al., 2001;DeMott et al., 2003b;Cziczo et al., 2004;Sassen et al., 2003). Ice residuals often contain mineral particles (Cziczo et al., 2013;Cziczo and Froyd, 2014;Twohy and Poellot, 2005). Mineral dust has high spatial and temporal variability, impacting atmospheric, oceanic, biological, terrestrial and human systems (Garrison et al., 2003;Gat et al., 2017;Jickells et al., 2005;Mahowald et al., 2014;Mazar et al., 2016;Middleton and Goudie, 2001). Each year, gigatones of dust are transported globally over long distances, dominating the atmospheric aerosol mass and aerosol optical depth (AOD) (Chiapello et al., 1999;Tegen and Fung, 1994;Ben-Ami et al., 2010;Prospero, 1999;Koren et al., 2006). Though the exact property of an aerosol that determines its ice nucleation ability remains unclear, it was consistently shown that the mineral composition plays an important role (Kanji et al., 2017), and that for a certain mineral type, larger particles are more effective heterogeneous INP than the small ones (Archuleta et al., 2005;Lüönd et al., 2010;Welti et al., 2019). Local surface features such as steps, cracks and cavities, a close match of the surface lattice with that of ice, or surface hydroxyl groups (Freedman, 2015;Marcolli, 2014;Zielke et al., 2015;Kiselev et al., 2017;Fletcher, 1969;Tunega et al., 2004;Anderson and Hallett, 1976;Pruppacher and Klett, 1997) are believed to be the responsible factors for the ice nucleation ability of mineral surfaces.

Natural mineral dust particles are often chemically similar but differ in their mineralogy (Engelbrecht et al., 2009), and the particles are often composed of a mixture of minerals (internally mixed), such as clays, quartz, feldspars and calcites (Claquin et al., 1999). Other common minerals are palygorskite, hematite, halite, gypsum, gibbsite and goethite (Ganor et al., 1991;Perlwitz et al., 2015;Kandler et al., 2007;Mahowald et al., 2014). The mineralogy of mineral dust is set by its source region and is considered to be an important factor that determines its freezing characteristics (Zimmermann et al., 2008;Augustin-Bauditz et al., 2014). Traditionally, clay minerals were thought to be responsible for atmospheric ice nucleation because they compose much of the dust fraction. However, using standard mineral particles, Atkinson et al. (2013) showed that K-feldspar is the most efficient type, and suggested that it could dominate atmospheric ice formation at relatively high

temperatures, above 258 K. This was further supported by measurements of natural mineral dust from desert surfaces
worldwide, where the importance of quartz mineral was also indicated (Boose et al., 2016b).
Airborne mineral dust (AMD) can experience chemical and physical modifications during its atmospheric transport that may
alter dust's ability to nucleate ice (Kanji et al., 2013). It was shown that atmospheric aging processes can change the size, the
morphology and the surface chemistry of the particles. For example, adsorption of organic components on AMD (Murphy et
al., 2006;DeMott et al., 2003a;Falkovich et al., 2004) or coatings of nitrates, chlorides and sulphates which enhance the
hygroscopicity of the particles (Krueger et al., 2004;Laskin et al., 2005;Li and Shao, 2009). Levin et al. (1996) found that
AMD particles transported over the Mediterranean Sea were often coated with sulphate and other soluble materials, which
affect clouds′ microphysical properties and can eventually result in enhanced ice nucleation. In addition, mineral dust carries
biological components, such as bacteria and fungi, which are known to have the ability to induce ice nucleation at relatively
high temperatures (Gat et al., 2017;Mazar et al., 2016;Pratt et al., 2009;O'Sullivan et al., 2016). Further modifications that can
occur during AMD atmospheric transport are the differentiation of size and mineralogy. These can occur due to gravitational
sedimentation, for example, where larger particles sediment faster than the smaller ones. Near source regions, dust samples
were richer in components that are more abundant in the coarse fractions, such as quartz and potassium feldspars, while in
remote locations, higher amount of clay minerals and sodium/calcium feldspar were observed (Murray et al., 2012;Schepanski,

79  2018).

While there are only few measurements of AMD close to source regions (Price et al., 2018;Boose et al., 2016a;Ardon-Dryer
and Levin, 2014;Schrod et al., 2017), parameterizations of ice formation in climate models are often based on the freezing
properties of natural dust or soil samples collected from deserts or standard dust particles  (Niemand et al., 2012;Connolly et
al., 2009;Ullrich et al., 2017;Atkinson et al., 2013;Broadley et al., 2012), that may not sufficiently represent AMD (Boose et
al., 2016b;Spichtinger and Cziczo, 2010). Natural dust samples showed higher ice nucleation ability than AMD samples,
possibly due to atmospheric processing of AMD that may lead to deactivation, and possibly due to laboratory processes, such
as milling or sieving, that were applied to the natural dust samples and may have enhanced its activity (Boose et al., 2016b).
In this study, we sampled airborne particles during dust events in the Eastern Mediterranean and investigated their ice
nucleation abilities. The Eastern Mediterranean is located in the strip of the world's main deserts, and experiences transport of
desert dust from different sources. The main source is the Sahara Desert in North Africa. It is estimated that about 100 million
tons of dust per year is lifted from the Sahara towards the Eastern Mediterranean, during late winter and spring (Ganor,
1994;Ganor and Mamane, 1982;Ganor et al., 2010). In autumn, local dust is transported, commonly from the Arabian Peninsula
and the Syrian Desert (Dayan et al., 1991;Ganor, 1994). The dust events are often associated with the regional Eastern
Mediterranean synoptic systems, such as winter lows and Red-Sea troughs (Ganor et al., 2010). Our sampling site was located
in Israel, where Saharan dust is transported over North Africa and/or the Mediterranean Sea, and Syrian and Arabian dust is
transported over land from the east (Ganor et al., 1991). These distinct sources and paths allow investigating the ice nucleation
properties of AMD with diverse origins and transport paths.
The ability of the collected particles to initiate immersion freezing was studied using the Weizmann Supercooled Droplets
Observation on a Microarray (WISDOM) instrument (Reicher et al., 2018), and one of the dust events was studied using the
Bielefeld Ice Nucleation ARraY (BINARY) instrument (Budke and Koop, 2015). We characterized the concentrations and the
density of ice nucleation active sites (INAS) of AMD in different size-classes for several dust cases, as well as combined recent
literature and available AMD data to understand how well AMD is represented in models based on recent parameterizations.

## 2 Data and Methods

### 2.1 Sampling

Airborne particles were sampled during six dust events in 2016 and 2017, detailed in Table 1. Sampling started when the
visibility reduced due to increasing concentrations of particulate matter (PM). The sampling site is located on a roof of a three-
story building in Rehovot, Israel (31.9N, 34.8E about 80m AMSL). The location is often impacted by mineral dust storms,
transported from nearby and distant geographical locations, mainly from the Sahara and Arabia deserts, and less frequently
from the Syrian Desert, depending on the season and the synoptic conditions (Dayan, 1986;Ganor et al., 2010;Kalderon-Asael
et al., 2009).
Particles were collected on polycarbonate filters (47mm cyclopore, 0.1 μm isopores; Whatmann), using the Micro-Orifice
Uniform Deposit Impactor (MOUDI) (model 110-R). The MOUDI is a 10-stage impactor with 18 μm cut-point inlet stage
followed by size segregating stages with cut points ($D_{50}$) between 0.056 and 10 μm in aerodynamic diameter (Marple et al.,
1991). The particles are collected on the different stages as function of their aerodynamic diameter. The collection efficiency
for each particle size is described in Marple et al. (1991). Sampling time ranged between 17 and 48 h with a 30 L min$^{-1}$ sample
flow rate, similarly to previous studies (Huffman et al., 2013;Mason et al., 2015).

### 2.2 Air mass Back Trajectories

Back trajectories were calculated by a Lagrangian method, using LAGRANTO 2.0 (Sprenger and Wernli, 2015). The
calculation of air mass trajectories was based on wind data from the European Centre for Medium-Range Weather Forecasts
ERA-Interim reanalysis (Dee et al., 2011), available every 6 h, at 1°×1° horizontal grid and 60 vertical hybrid levels. For each
6-h time step during each event, 72 h back trajectories were calculated, from all available data grid points with pressure larger
than 850 hPa, resulting in 11 trajectories, which end their path in the lower troposphere for each calculation. In a second step,
the Eulerian densities of the resulting trajectories were computed by gridding the trajectories for each event, smoothed by using
a radius of 100 km and interpolated to 1 h. Finally, the trajectory density was summed over the entire event duration and
normalized by the maximum trajectory count.
**2.3 Dust Column Mass Density Maps**
Time averaged maps of dust column mass density (hourly 0.5°×0.625°) reanalysis data were obtained from the Modern-Era
Retrospective analysis for Research and Applications (MERRA-2). Maps were produced using NASA's Global Modeling and
Assimilation Office (GMAO) (Gelaro et al., 2017), for a period of up to 72 h prior to the sampled event.
**2.4 Particulate Matter Data**
Particulate matter mass data were obtained from the Israeli Ministry of Environment website. Concentrations of particles with
aerodynamic diameters smaller than 10 μm ($PM_{10}$) were measured in the Rehovot station, located about 1 km from our
sampling site. The 5-minutes mean data was used to calculate peak and mean concentrations of the sampled dust events.
**2.5 Particle Number-Size and Surface Area distributions**
Particle size distribution and concentrations between 0.25 and 32 μm was measured on site by an optical particle counter (OPC;
GRIMM Technologies model 1.109), in parallel to the MOUDI sampling. In order to estimate the total surface area that was
collected on the different stages, we assumed that the particles are spheres and used the diameter of the GRIMM midpoint of
the different GRIMM's channels as the particles' diameter.
**2.6 Conversion of GRIMM channels to MOUDI stages**
To determine the total surface area collected on MOUDI's filter, a conversion matrix between the GRIMM channels and the
MOUDI stages was applied. The conversion was based on the particle collection efficiency curves of the MOUDI and inter-
stage particle losses reported in Marple et al. (1991). Figure 1 demonstrates the fraction of particles that are collected on the
stages based on their aerodynamic diameter. Freezing analyses focused on stage #2 ($D_{50}$ = 5.6 μm), stage #3 ($D_{50}$ = 3.2 μm),
stage #4 ($D_{50}$ = 1.8 μm), stage #5 ($D_{50}$ = 1.0 μm), stage #6 ($D_{50}$ = 0.6 μm) and stage #7 ($D_{50}$ = 0.3 μm). For example, most of
the particles with an optical diameter > 8.5 μm will be collected on stage #2 ($D_{50}$=5.6 μm), whereas all the particles with an
optical diameter > 17.5 μm are assumed to be collected on former stages (inlet and stage #1). In some cases, particles in a
certain size are likely to impact on two different MOUDI stages. For example, a small fraction of particles with 0.5 μm optical
diameter are collected on stage #5 ($D_{50}$=0.6 μm), and most of them impact on stage #6 ($D_{50}$=0.3 μm). The initial particle
concentration that was used is the accumulated sum of all particles for the entire sampling period.
**2.7 Ice Freezing Experiments and Quantification**
**2.7.1 WISDOM**
Immersion freezing activity of the sampled ambient mineral dust was measured using suspensions of the collected particles
that were extracted from the filters by dry sonication (VialTweeter, model UP200St; Hielcher). This type of sonication method
is more effective than the ultrasonic bath in which most of the energy is dissipates in the surrounding water. A quarter filter

was inserted into a 1.5 ml Eppendorf vial with 0.3 ml deionized water, and sonicated in three 30 s cycles, to avoid heating produced during intense sonication. The suspension was immediately used for droplet production and freezing experiments in WISDOM as detailed in Reicher et al. (2018). Briefly, an array of 0.5 nL monodispersed droplets (~100 μm diameter, suspended in an oil mixture) was generated in a microfluidic device that was cooled by a commercial cooling stage (THMS600, Linkam) under a microscope (BX-51 with 10X magnification, Olympus) coupled to a CCD camera. The device was first cooled at a faster constant rate of 10 K min$^{-1}$ from room temperature to 263 K, since freezing events were not expected and indeed were never observed in that temperature range. Then a constant cooling rate of 1 K min$^{-1}$ was used until all the droplets froze. The temperature uncertainty was ± 0.3 K, based on error propagation between the calibrated droplet temperature and the uncertainty of the temperature sensor that is located in the cooling stage (see Reicher et al. (2018) for more details).

### 2.7.2 BINARY

The Bielefeld Ice Nucleation ARraY (BINARY) is an optical freezing array of droplets pipetted on a hydrophobic substrate in separated sealed compartments and cooled in a Linkam cooling stage (LTS120) (Budke and Koop, 2015). In the present study an array of 64 droplets of 0.6 μL was employed. Suspensions were prepared by extracting a quarter filter in 1.5 ml of double-distilled water (that is, 5 times more diluted than WISDOM suspensions), using a bath sonicator (Elma Transsonic Digital, TP 670/H) for 30 min.  The bath temperature increased during sonication from about 288 to 308 K. The obtained suspensions were used directly and further diluted (1:10) for another set of measurements with reduced surface area of the particles in the droplets. For the freezing experiments, the droplets were cooled at a rate of 1 K min$^{-1}$. Temperature uncertainty was ±0.3 K.

### 2.7.3 Quantification of Freezing Properties

The cumulative concentration of INP present in a volume of solvent, $V$, at temperature $T$, was derived using the fraction of frozen droplets ($f_{ice}(T)$), that was obtained directly from the freezing experiments (Vali, 1971):

$$K(T) = \frac{-ln(1-f_{ice}(T))}{V} \quad [cm^{-3} \; of \; water] \qquad (1).$$

For control experiments, a quarter of blank filter was immersed in pure water, similarly to freezing experiments of the airborne samples, and the concentration of the background impurities ($K_{imp}(T)$) were subtracted from the concentrations that were detected for airborne samples.

The atmospheric concentrations of INP per unit volume of air as a function of temperature, $INP(T)$, were determined by incorporating the sampling and solvent parameters into Eq. 2 (Hader et al., 2014):

$$INP(T) = \left(K(T) - K_{imp}(T)\right)\frac{V_{\text{solvent}}}{f \cdot V_{\text{air}}} \quad [L^{-1} \; air] \qquad (2),$$

where $V_{solvent}$ is the volume of the water used for extraction, $V_{air}$ is the total sampled air volume, and $f$ is the fraction of filter that was used in the extraction.

For comparison of ice nucleation activity of the different dust events, the INP concentration in the liquid was converted to the
number of active sites per unit surface area of INPs, i.e., the surface density of sites $n_s$ active above temperature, $T$ (Vali,

187  1971):

$n_s(T) = \frac{-\ln(1-f_{ice}(T))}{A}$  $[m^{-2}]$                                        (3),
where $A$ is the surface area immersed in a single droplet of the experiment, based on the total surface area of particles in the
suspension.
**2.8 Scanning Electron Microscopy**
A quarter of selected filters were coated with Iridium for analysing the chemical composition of airborne particles using a
scanning electron microscope (SEM; Supra 55VP, LEO) equipped with an Energy-dispersive X-ray spectroscopy (EDX)
detector for elemental microanalysis. The analysis was done at a voltage of 5 kV using the Quantax software (Bruker).
**3 Results and Discussion**
**3.1 Air mass Back Trajectories and the Origin of the Dust Storms**
The density of air mass back trajectories for 72-h period prior to the sampling for all events are shown in Figure 2. The sampling
site and the surrounding main deserts are shown as well. During the sampled events, the air mass trajectories were diverse. In
some cases, the air masses travelled directly to the sampling site from the source region, while in other cases, they travelled
longer distance. In most events, the air mass had either easterly or westerly component, and were often concentrated in the
same geographical area.
The dust origins were identified based on back trajectory analysis, integrated with reanalysis data of remote sensing of
atmospheric dust. We followed the dust mass concentration prior to the sampling period, as detailed in Figure S1. Locations
that contained high levels of suspended dust and overlapped with the air mass trajectories were identified as the possible
sources of dust. The green contours in Figure 2 represent the assigned dust origin for each sampled event based on the reanalysis
data. Note that in two events, there was no overlap between the dust origin and air mass trajectories. These events will be
further discussed below. Two events, denoted by SDS1 and SDS2, originated in North Sahara Desert. The source of SDS1 was
near the border of Egypt and Libya, and the source of SDS2 was in Egypt, east of SDS1. The dust travelled over the
Mediterranean Sea and was potentially affected by the marine environment, possibly obtaining a sea salt or anthropogenic
sulfate coating (Levin et al., 1996). Two other events, denoted by SyDS1 and SyDS2, originated from the Syrian Desert, from
western Iraq and southern Syria. Compared to the Saharan events, the dust mass density in the Syrian Desert events was
relatively low.
Another event was defined as a "mixed dust" event (MDS), because it was more complicated and included contributions of
different sources: the analysis indicates that there is one possible dust origin east of the sampling site in the Syrian Desert, and

another one southwest of the sampling site in the Sahara Desert. However, the air mass trajectories did not overlap with the Saharan dust origin, but indicated that the air mass was transported from the Red Sea. Further analysis of the air mass trajectories prior to the sampling period in the Red Sea showed that both Sahara and Arabia dusts were transported to the Red Sea (see the supplementary part, Figure S2(a)). Another event did not show overlap between the air mass trajectories and the dust origin. Further analysis of air mass back trajectories in the days prior to the sampling period showed that dust was transported to the Mediterranean Sea from the region of Libya in the Sahara Desert, towards Turkey, and was deflected eastward by westerly winds to the sampling site (see the supplementary part, Figure S2(b)). The dusty air masses rapidly cleared up, and relatively non-dusty air masses arrived at the sampling site, as inferred from $PM_{10}$ concentrations and the OPC size distributions, see section 3.2. This event was defined as "clean and Saharan dust storm" and denoted by CSDS. Table 1 summarizes the sampled events, their sampling periods, and the peak and mean $PM_{10}$ concentrations during sampling. Peak values ranged from 67 $\mu g\ m^{-3}$ in CSDS and 132 $\mu g\ m^{-3}$ in SyDS1, to 717 $\mu g\ m^{-3}$ in SDS2, which was the strongest dust event in this study. In SDS1, MDS and SyDS2, the values ranged between ~300 to 400 $\mu g\ m^{-3}$. When comparing the mean $PM_{10}$ concentrations during the entire sampling periods, CSDS was categorized as a non-dusty event, with the lowest concentrations of 30±13 $\mu g\ m^{-3}$, i.e. below the threshold of 42 $\mu g\ m^{-3}$ for dusty conditions (Krasnov et al., 2014). The mean values in the rest of the events ranged from 76 to 206 $\mu g\ m^{-3}$, and were therefore categorized as dust storms.

**3.2 Particle Number-Size Distributions**

Figure 3(a) describes the mean particle number-size distributions of sampled air during the dust events, as was detected by the GRIMM OPC. The lowest channel of the GRIMM includes particles that are larger than 0.25 $\mu m$. This channel possibly underestimates the total particle count since the counting efficiency is less than 100%.

The number-size distributions had similar patterns in all the events. The highest particle number concentrations were in the submicron size range, decreasing towards larger particles. Events SDS1, SDS2, and MDS had a rather similar particle concentration distribution. Event SyDS1 showed similar particle concentrations in the submicron range, but the particle concentrations in the supermicron range were about an order of magnitude lower, which was also apparent in the $PM_{10}$ data. CSDS, a predominantly non-dusty event, had the lowest particle concentrations in comparison to the rest of the sampled events, as was also indicated by the $PM_{10}$ data. In the SyDS2 event, exceptionally high concentrations in the supermicron range above 3 $\mu m$ were observed, and the peak extended towards larger particle sizes, combined with relatively high particle concentrations. Note that prior to and during this event, a series of biomass burning events occurred in Israel extending to about100 km north and 50 km east of the sampling site. Therefore, this peak may include also contributions from biomass burning particles. This is further supported by the SEM-EDX analysis of the filters from this event, which in comparison with the other events, contained super-aggregates in the supermicron range, typically observed in biomass burning emissions (Chakrabarty et al., 2014), with distinct morphologies and elemental composition (shown in the supplementary part in Figure S3).

The surface-area-size distributions shown in Figure 3(b) compare the contribution of supermicron and submicron particles to
the available ambient surface area. Ice nucleation initiated on the surface of the particles, and therefore, their surface area
concentration is an important parameter in addition to number concentrations. Here it is clearly seen that the potential
contribution of the supermicron particles to the ice nucleation may be significant when compared to the submicron particles,
although their number concentrations were up to two orders of magnitude lower.

**3.3 Airborne INP Concentrations**

The cumulative INP concentration spectra for the six dust events are shown in Figure 4. In each event, different particle size
classes are marked by different color. Freezing was observed between 255 and 238 K, and the INP concentrations spanned
four orders of magnitude from $10^{-1}$ to $10^{3}$ $L^{-1}$ of air.
A particle size dependence of the freezing temperature and INP concentration was observed. Larger particles froze at warmer
temperatures with higher number of INPs. The variation between the six size-classes ranged from 1 to 2 orders of magnitude,
and in some cases the smallest particles had similar behavior to the large ones. For example, in event SDS2, size-classes
$D_{50}$=0.6 μm and $D_{50}$=0.3 μm were less ice-active than the rest of the size-classes, while in MDS, all size classes showed similar
activity. As an exception, event SyDS2 showed a weaker size dependence in comparison to the other dust events, and in some
size-classes, lower INP concentrations. In comparison, in the relatively non-dusty event CSDS, the variability between the
different size classes was higher, especially at lower temperatures. In Figure 5, similarly to Figure 4, INP concentrations are
presented, but arranged according to the different size classes. The variability within each size class was relatively high and
spans over 2 orders of magnitude; for example, at size class $D_{50}$=0.3 μm near 245 K, INP concentration ranged from about 1
to almost $10^{2}$ $L^{-1}$ of air. It is clearly seen that INP concentrations in dusty conditions (SDS1, SDS2, MDS and SyDS1) were
higher than in non-dusty conditions (CSDS) for the supermicron range, but similar in the submicron range. Previous studies
also pointed out the significant contribution of supermicron particles to the INP population. Mason et al. (2016) studied the
immersion freezing abilities of airborne particles in North America and Europe, and found that supermicron particles
dominated the freezing, especially at relatively high temperature (258 K). Recent measurements in a coastal tropical site
conducted by Ladino et al. (2019) also found high concentrations of INPs at relatively high temperatures (> 258 K) due to
supermicron particles. In these studies, however, mineral dust is not expected to dominate the samples, and bioaerosol particles
are thought to dominate the freezing at the high temperatures (> 258 K). At lower temperatures (below 253 K), Ladino et al.
(2019) suggested that mineral dust dominated the freezing. Moreover, DeMott et al. (2010) found that INP concentrations are
correlated with particles > 0.5 $\mu$m. Other studies, such as Rosinski et al. (1986) and Huffman et al. (2013), also found that
supermicron particles were responsible for most of the INP population in some cases, while when changing the freezing mode
that was analysed or the measurement meteorological conditions, their contribution was reduced. Vali (1966) in contrast, found
that submicron particles dominate freezing in hail melt samples.

## 3.4 Size-Dependence of Ice Active Site Density ($n_s(T)$)

Figure 6 presents the $n_s(T)$ curves for the different dust events spanning a range of $10^6$ m$^{-2}$ at 253K to $10^{11}$ m$^{-2}$ at 238 K. In
general, $n_s(T)$ increased with particle size. The highest $n_s$ values were observed in the supermicron range D$_{50}$=5.6 µm,
followed by D$_{50}$=3.2, 1.8 and 1.0 µm. The activity of the latter three classes was similar within measurement uncertainties. In
the submicron range, stages D$_{50}$=0.6 and 0.3 µm, the $n_s(T)$ values were lower than in the supermicron range and showed
higher variability between the different events, except for the MDS event, that had similar activity in the submicron and the
supermicron range. While INP concentrations may generally vary due to experimental parameters, such as particle
concentration in the droplet or droplet size, $n_s(T)$ accounts for these differences since it is normalized by the total surface area
of particles immersed in the droplet. Therefore, the effect of particle size diminishes using the $n_s(T)$ curves, if the particles'
ice-nucleation ability is indeed similar. Hence, the analysis presented in Figure 6 indicates that the supermicron particles are
better INP than the submicron ones, implying they have more active sites or/and active sites that nucleate ice at higher
temperatures.
Figure 7 displays the same $n_s(T)$ curves as Figure 6, but now arranged according to the different size-classes. It is observed
that in the supermicron range, all $n_s(T)$ curves from the different events merge (with the exception of SyDS2) suggesting that
freezing was dominated by a common component. While the freezing activity decreases with decreasing particle size, the
shape of the curves is preserved, suggesting that the abundance of this common component decreases with particle size. One
possible explanation for this observation may be mineralogy segregation, known to occur with particle size: larger particles
contain more primary minerals, such as K-feldspar, whereas smaller particles contain more secondary minerals, such as clays
and quartz that are common in all particle sizes (Perlwitz et al., 2015;Claquin et al., 1999). Therefore, the reduced activity in
the submicron range and the higher variability between the dust events, especially at D$_{50}$=0.3 µm, may be attributed to a
different mineralogical composition of the particles, or to the lack of the important ice-inducing component. Alternatively, it
is also possible that the submicron particles are mixed with other particle types, that are more common in this size range, such
as urban pollution (Weijun et al., 2016), and therefore freezing may not be dominated exclusively by mineral dust. Moreover,
due to their larger surface-to-volume ratio, submicron particles are more sensitive to atmospheric processing than supermicron
particles, which can lead to further deactivation of their ice active sites (Boose et al., 2016a). These considerations may explain
the variability in the activity between different events. For example, we propose that the passage of SDS1 and SDS2 over the
Mediterranean Sea can contribute to their reduced activity in the submicron range, while for the MDS event, a shorter and
relatively direct transport path resulted in less atmospheric processing. Although speculative, these considerations may
possibly explain why the freezing activity of submicron particles converged with those of the supermicron particles, but we
acknowledge that further measurements are needed to confirm these suggestions.
In Figure 7, we also compare a few relevant $n_s(T)$ curves of standard minerals, as derived by Atkinson et al. (2013) and
Niedermeier et al. (2015), together with our measured $n_s(T)$ curves. The standard curves of K-, Na/Ca-feldspar and quartz
were scaled to the estimated fraction of these minerals in AMD (see Table S1), and are typically used for prediction of AMD
ice nucleating activity. A good agreement of the absolute $n_s$ values was observed in the relevant temperature range, and the
slopes of the curves were similar to those of the feldspars, especially for the supermicron range. A good agreement was also
observed with the standard $n_s(T)$ curve of quartz, suggesting that it contributes to freezing of the submicron particles in the
lower temperature range. Note that the standard $n_s(T)$ curves of clay minerals and calcite were not plotted here despite their
large abundance in AMD, because there was no overlap with the ice nucleation activity in this study. Only the freezing activity
of the largest particles ($D_{50}$=5.6 μm) overlapped with the K-feldspar prediction of Atkinson et al. (2013), indicating that this
prediction possibly overestimates the freezing activity of the entire size distribution of AMD. For the particles in the size range
of 3.2 < $D_{50}$ < 1.0 μm, there is an overlap in activity with the K-feldspar prediction of Niedermeier et al. (2015) and Na/Ca-
feldspar of Atkinson et al. (2013). However, in all cases, the feldspars predictions overestimate the freezing activity of AMD
in the submicron range.
The $n_s(T)$ curves of SyDS2 displays moderate slopes and lower IN activity in comparison with the other dust events, in all
size classes, except for the smallest particles with $D_{50}$=0.3 μm. As was already mentioned, these particles were most likely
mixed with smoke particles from biomass-burning events that occurred during the same period, and the filters from this event
were covered with super-aggregate particles in the supermicron size, rich with potassium, similar to particles seen in other
biomass burning events (Chakrabarty et al., 2014).

## 3.5 Comparison of WISDOM and BINARY measurements for event CSDS

A complementary analysis for the CSDS event using BINARY is shown in Figure 8. BINARY probes droplets with larger
volumes and, thus, it is more sensitive to less common ice-nucleating sites that may not show a signal in WISDOM. In the
BINARY experiments, two suspensions were tested, with different dilution factors, for extending our sensitivity. The higher
total dust surface area per droplet sample that was investigated in the BINARY experiments, yellow markers in Figure 8,
demonstrates the warmest freezing temperatures, ranging from 255 to 246 K, and the $n_s(T)$ values ranged from $10^6$ to $10^9$ m-
$^2$. The 1:10 diluted samples (purple markers) showed freezing at lower temperatures, ranging from about 251 to 244 K, with
higher $n_s(T)$ values ranging from $10^8$ to $10^{11}$ m$^{-2}$. In some of the dilute cases of the BINARY experiments, the data were at
the limit of the background impurities (see supplementary part, Figure S5). In order to include only data that are significantly
different from the background, a criterion was set, in which only those data points that are larger by at least two standard
deviations than the mean background impurities were further considered in Figure 8. If data were below that threshold, they
were considered as not significant and thus were removed (e.g., the data of the $D_{50}$=0.6 and 0.3 μm for the diluted BINARY
samples).
Figure 8 shows a very good agreement between the BINARY and WISDOM data, because the $n_s(T)$ curves merged nicely
onto each other for each size-class. Whereas BINARY was more sensitive than WISDOM to the warmer and relatively rare
active sites, WISDOM detected the more common active sites in the low temperature range. Overall, the dependence of the
freezing activity temperature range on the immersed surface area per droplet is well demonstrated here, where a reduction in
the surface area of the different experiments (WISDOM < BINARY diluted < BINARY) decreased the probability to observe
freezing at the higher temperatures. This was also demonstrated previously in studies of standard mineral dust (Broadley et al.,
2012;Marcolli et al., 2007;Reicher et al., 2018). Overall, the data shown in Figure 8 indicate the added value when using
experimental techniques of different sensitivity for the purpose of measuring the concentration and active site density of INP
in field studies (e.g., Atkinson et al. (2013); Chen et al. (2018);Harrison et al. (2018)).

## 3.6 Comparison of Super- and Submicron ranges with AMD Measurements and Predictions

The particle surface area that was used to derive $n_s(T)$ represents the total airborne particles that were collected for each
sample, regardless of particle composition. When mineral dust dominated the composition, as in a dust event case (see for
example Figure S4 in the supplementary part), we treat $n_s(T)$ as representative for AMD freezing. Figure 9(a) compiles the
$n_s(T)$ results of AMD from a few recent studies that focused on airborne particles (albeit not size-selected) during dust events.
Results from our current study, excluding the events SyDS2 and CSDS that were not dominated by AMD, are presented
alongside those of Price et al. (2018) and Boose et al. (2016b). Price et al. (2018) collected airborne particles in flights west of
the Sahara Desert over the tropical Atlantic at altitudes of up to 3.5 km. Boose et al. (2016b) analysed airborne particles which
were deposited in the Eastern Mediterranean region in Egypt, Cyprus and the Peloponnese (Greece) during dust events. Boose
et al. (2016b) also sampled airborne particles during dust events over Tenerife, off West Africa. In addition, we present
measurements which were also conducted in the Eastern Mediterranean region in Cyprus. Schrod et al. (2017) measured INP
in the lower troposphere using an unmanned aircraft system and Gong et al. (2019) measured INP at ground level. Both studies
measured the immersion freezing of the sampled particles during different atmospheric conditions that included few dust
plumes from the Sahara. Note that here we present only immersion/condensation freezing measurements by Schrod et al.
(2017) and not the entire data . Also note that the presented data is not necessarily dominated by mineral dust, in contrast to
the current study or to Price et al. (2018) and Boose et al. (2016b). The specific cases where the samples were taken during
passage of dust plumes and are possibly dominated by mineral dust are marked in Figure 9(a) in green for Schrod et al. (2017)
and cyan for Gong et al. (2019). The supermicron data presented in this paper is about 1 to 2 orders of magnitude higher, while
our submicron data is in relatively good agreement with Schrod et al. (2017), except for the lowest temperature (243 K) points
where 1 to 3 orders of magnitude differences were observed. The Gong et al. (2019) data are lower in 1 to 3 orders of magnitude
but there is some overlapping with this study and with Price et al. (2018).
This compilation of the data that was dominated by mineral dust (i.e., this study, Price et al. (2018) and Boose et al. (2016b))
shows that $n_s(T)$ curves from the different studies exhibit great similarities over a wide range of temperatures (236 - 265 K)
for dust from different locations and geographic sources, with varying atmospheric paths and altitudes. This similarity may
have significant implications for modelling ice nucleation activity by AMD, since it suggests that parameterizations can be
simplified, for example by neglecting the complication of accounting for mineralogy of different geographical sources. Due
to the different behaviour of submicron and supermicron particles, we also suggest that accounting for the particle size class
will improve the prediction of ice cloud formation. For that purpose, we derived two basic parameterizations (Eq.4), for
supermicron and submicron particles, based on the combined AMD data (including data from this study, Price et al. (2018)
and Boose et al. (2016b), and excluding SyDS2), which cover a wide range of temperatures, and spread more than 5 orders of
magnitudes in $n_s(T)$ values. These parameterizations are the best mathematical fit for a Hill-type equation, which is normally
used for fitting S-shaped data as they are observed in this compilation:
$$n_s(T) = exp[y_0 + a/(b + \exp[(T - 248)/c])]\ [m^{-2}] \qquad\qquad (4)$$
where the coefficients (95% confidence bounds) for supermicron range particles are set to:
$y_0 = 11.47\ (10.97, 11.98), a = 24.00\ (22.01, 25.99), b = 1.53\ (1.35, 1.70), and\ c = 4.54\ (4.06, 5.02),$
$T \in [236K, 266K]\ (R^2 = 0.93)$.
and for submicron range:
$y0 = 9.48\ (8.19, 10.76), a = 23.00\ (20.23, 25.77), b = 1.34\ (1.10, 1.57), and\ c = 7.38\ (5.84, 8.92),$
$T \in [238K, 266K]\ (R^2 = 0.93)$.
Parameterizations for each individual size class can be found in Table S2 in the supplementary part.
In Figure 9(b), the parameterizations derived here are presented next to the recent parameterizations of ice nucleation of desert
dust by Ullrich et al. (2017) and Niemand et al. (2012). These parametrizations are based predominantly on natural surface-
collected dust samples, but also contained one sample of AMD from Israel, and agrees within an order of magnitude with our
supermicron data in the low-temperature range (243 - 247 K), but overpredicts $n_s(T)$ by more than an order of magnitude
when compared to our submicron data and to the Price et al. (2018) data at warmer temperatures (247-259 K). This emphasizes
that AMD ice nucleation may not be correctly represented when based on desert dust sampled from the surface, consistent
with the conclusions of Boose et al. (2016b) who showed that the average freezing activity of AMD is reduced when compared
to the activity of surface-collected desert dust. K-feldspar parameterizations by Atkinson et al. (2013) and Niedermeier et al.
(2015) are also shown here, and as mentioned before, overpredicts the freezing activity of AMD at temperatures lower than
about 255 K.
**4 Conclusions**
We characterized the INP activity of particles collected during several mineral dust events in the Eastern Mediterranean. Dust
from the Sahara Desert, the major source for atmospheric dust, together with dust from the Arabian and Syrian deserts were
included. Six size classes were studied that cover both the super– and submicron size ranges. The INP concentrations ranged
from $10^{-1}$ L$^{-1}$ of air in the relatively weak dust events to $10^3$ L$^{-1}$ of air in the strongest event. The $n_s$ values ranged from $10^6$ to
$10^{11}$ m$^{-2}$ in the temperature range of 238 – 255 K. A size dependence was observed, both in the INP concentration and in $n_s$
values. Larger particles were more active INP, exhibited higher INP concentrations and a higher number of nucleating sites
per surface area at higher temperatures. Comparison between freezing results of WISDOM with BINARY showed good
agreement, strengthened previous studies that observed how the freezing activity could depend on technical properties and
limitations of the used instrumentation, and therefore emphasize the importance of using complementary instruments.
The dust events studied here represent a range of dust loads, different dust origins and atmospheric paths. Yet, the supermicron
particles in these events exhibited similar freezing abilities. This may indicate that there is a unique component that is
responsible for freezing activity, as was previously suggested (Atkinson et al., 2013;Boose et al., 2016b;Kaufmann et al.,
2016;Price et al., 2018). Our measurements showed that the activity of the supermicron particles was in the range of standard
particles of feldspar mineral, and that the activity of the submicron particles was in the range of standard quartz. Therefore,
we suggest that these may be the two most important components that dominate the freezing by atmospheric mineral dust
(AMD), and therefore may be important for heterogeneous ice nucleation in atmospheric clouds. The submicron particles
showed higher variability between events, possibly due to different composition of the particles or higher sensitivity to
atmospheric processing during long-range transport. In general, supermicron particles contributed the most to the INP
concentration, in agreement with other previous studies (Mason et al., 2016;Huffman et al., 2013;Ladino et al., 2019).
However, our current study is probably the only case where mineral dust dominated the samples. Nevertheless, all of these
studies highlight the importance of the supermicron size class of AMD for atmospheric ice nucleation.
Mineral dust is important both on a regional scale, near its source region, and on a global scale, since it remains ice-active even
after long transport in the atmosphere and thus over considerable distances (DeMott et al., 2003b;Chou et al., 2011). With the
distance from the dust source, supermicron particles will settle, and submicron particles may then dominate ice nucleation on
the global scale (Ryder et al., 2013;Murray et al., 2012). However, recent airborne measurements found coarse and giant
particles in the vicinity and also far from source regions (Ryder et al., 2018). Therefore, including the particle size class in INP
parameterizations can improve predictions of ice formation in clouds. Moreover, information on airborne INP size distributions
may be helpful in identifying the dominant INP sources (Mason et al., 2016). The overprediction of AMD freezing ability
demonstrated in this study, by the Atkinson et al. (2013), Niedermeier et al. (2015);Niemand et al. (2012) and Ullrich et al.
(2017) parameterizations, especially for submicron particles, emphasizes the importance of future studies to better quantify
the changes in the ice-nucleating properties of AMD by atmospheric processing.


**Data availability**. Data are available upon request to the first author.

**Author contributions.** NR and YR designed the experiments, carried out the field measurements, conducted freezing
experiments in WISDOM, and wrote the paper. CB, LE and TK designed and performed freezing experiments in BINARY.
SRR performed backtrajectory analyses. NR and IKA performed the chemical analyses of filters. All authors contributed to
the discussion and analysis of data and the writing of the manuscript.

**Competing interests.** The authors declare that they have no conflict of interest.
**Acknowledgments**
This study was partially funded by the Israel Science Foundation (grant #236/16). The authors are grateful for funding by the
German Research Foundation (DFG) through the research unit FOR 1525 (INUIT) under KO 2944/2-2 for C.B. and T.K., and
a Mercator Fellowship for Y.R., and acknowledge the support from The Helen Kimmel Center for Planetary Sciences, and the
de Botton Center for Marine Sciences. Analyses and visualizations of MERRA data in this study were produced with the
Giovanni online data system, developed and maintained by the NASA GES DISC (http://giovanni.sci.gsfc.nasa.gov/Giovanni),
PM$_{10}$ data is available from the Israel Ministry of Environmental Protection website
(http://www.svivaaqm.net/Default.rtl.aspx). Other data used in this study can be retrieved from osf.io/gpuqt.

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

**Table 1: Summary of the investigated dust storm events. The events are denoted by their geographic origin: Saharan dust storms (SDS), Syrian dust storm (SyDS), mixed contribution of the two (MDS), and mix of dust event with a free-dust period (CSDS).**

| Event | Start [UTC] | Sampling period [hour] | PM10 peak [μg m$^{-3}$] | PM10 mean [μg m$^{-3}$] | Freezing analysis technique |
|---|---|---|---|---|---|
| **SyDS1** 19 April 2016 | 07:30 | 25 | 132 | 76±20 | WISDOM |
| **CSDS** 27 April 2016 | 07:30 | 24 | 67 | 30±13 | WISDOM, BINARY |
| **SyDS2** 23 November 2016 | 15:30 | 18 | 332 | 184±68 | WISDOM |
| **SDS1** 09 March 2017 | 11:00 | 48 | 387 | 96±66 | WISDOM |
| **SDS2** 12 March 2017 | 12:00 | 24 | 717 | 206±120 | WISDOM |
| **MDS** 12 April 2017 | 13:30 | 25 | 409 | 141±106 | WISDOM |



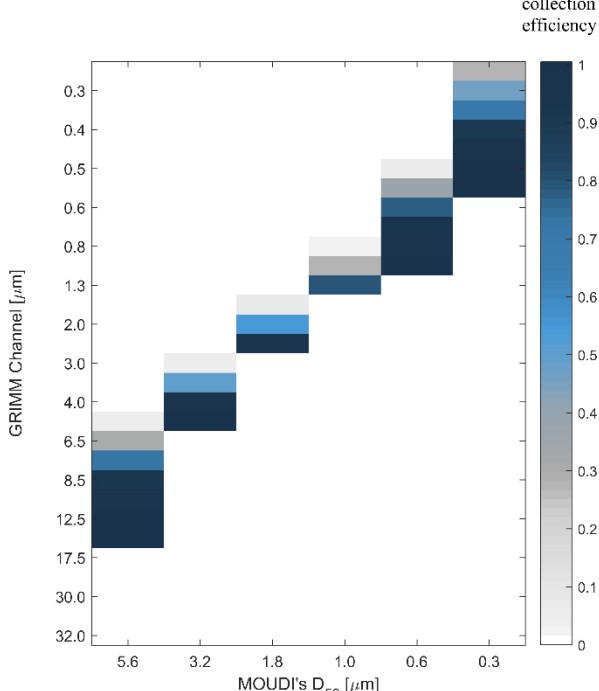


**Figure 1: A conversion matrix of GRIMM channels to MOUDI stages. The conversion was based on collection efficiency curves from Marple et al. (1991). The color shades represent the fraction of particles of a specific GRIMM channel to be impacted on a specific MOUDI stage.**

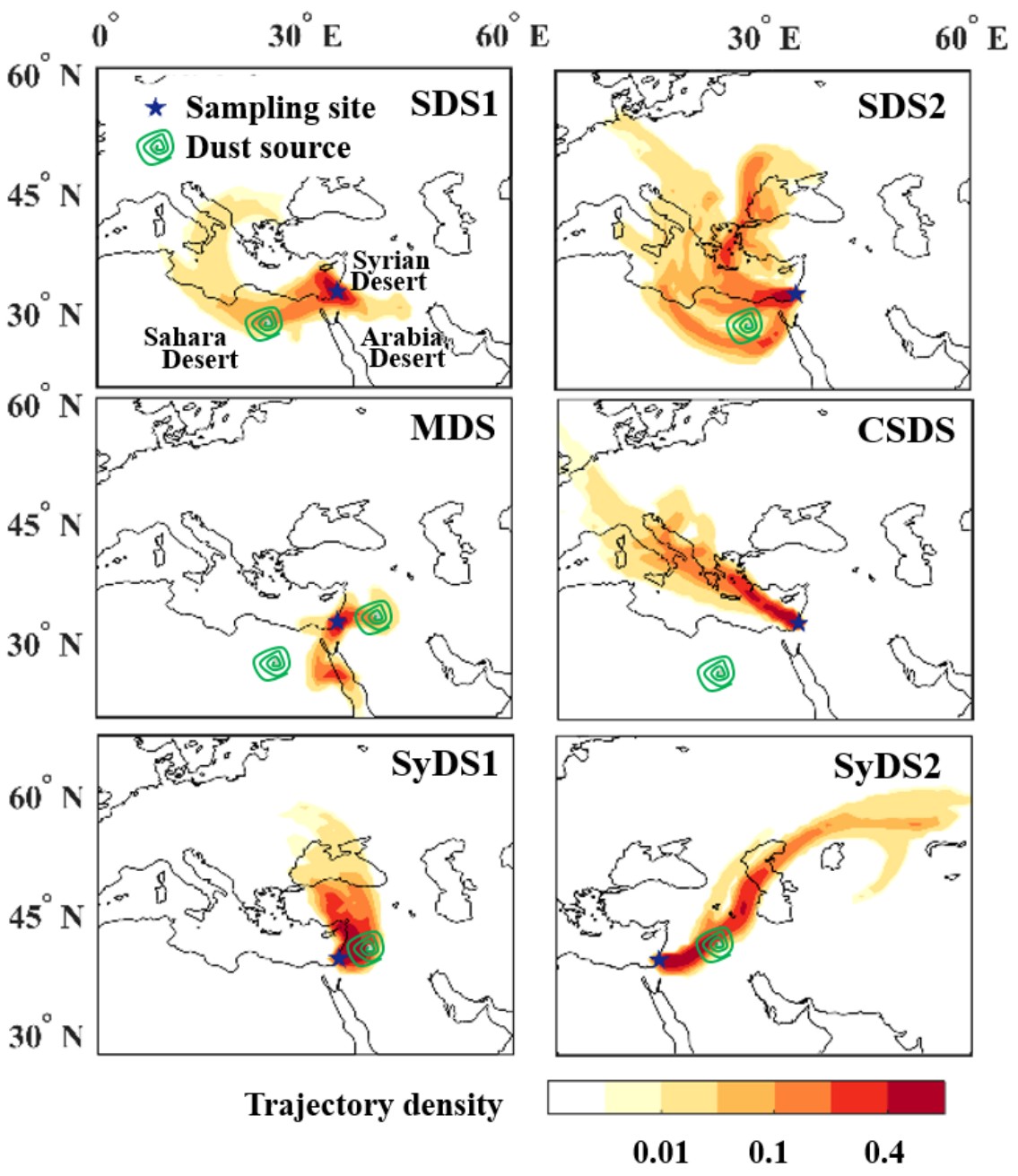


Figure 2: Air mass origin and atmospheric paths of the dust events. Colors represent the density of 72-h backward air mass trajectories (normalized to the total trajectory counts). The green contours represent the geographic locations where a high mass of the dust occurred during the air mass transition, which defined as the potential origin of the dust. Abbreviations in the top right of each panel indicate the particular dust event.


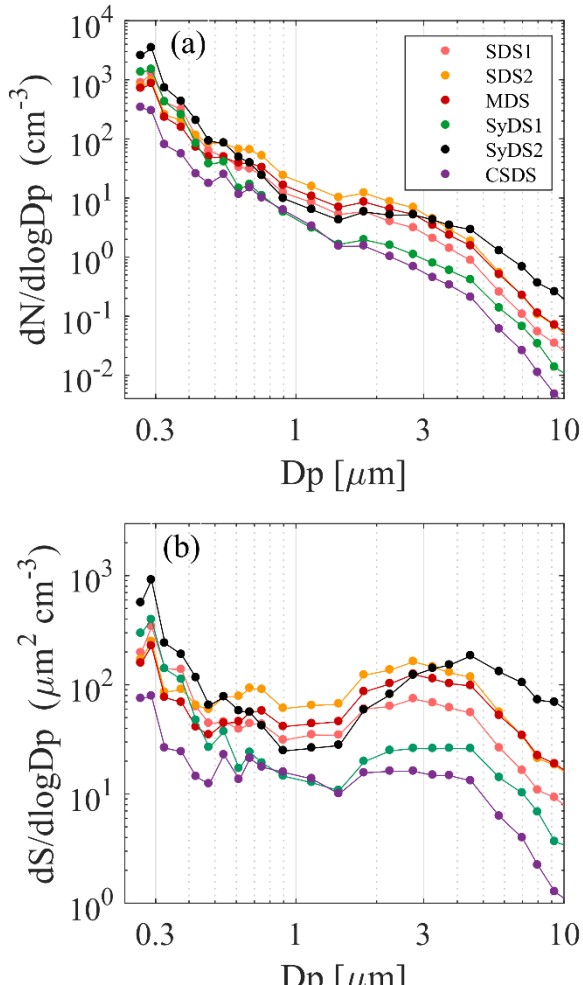


**Figure 3: Particle size distributions. Particle number size (a) and surface area size (b) distributions averaged over the entire sampling periods of the events as monitored by GRIMM OPC during the studied events. Dp is the diameter of the particles and set at the center of each GRIMM channel.**






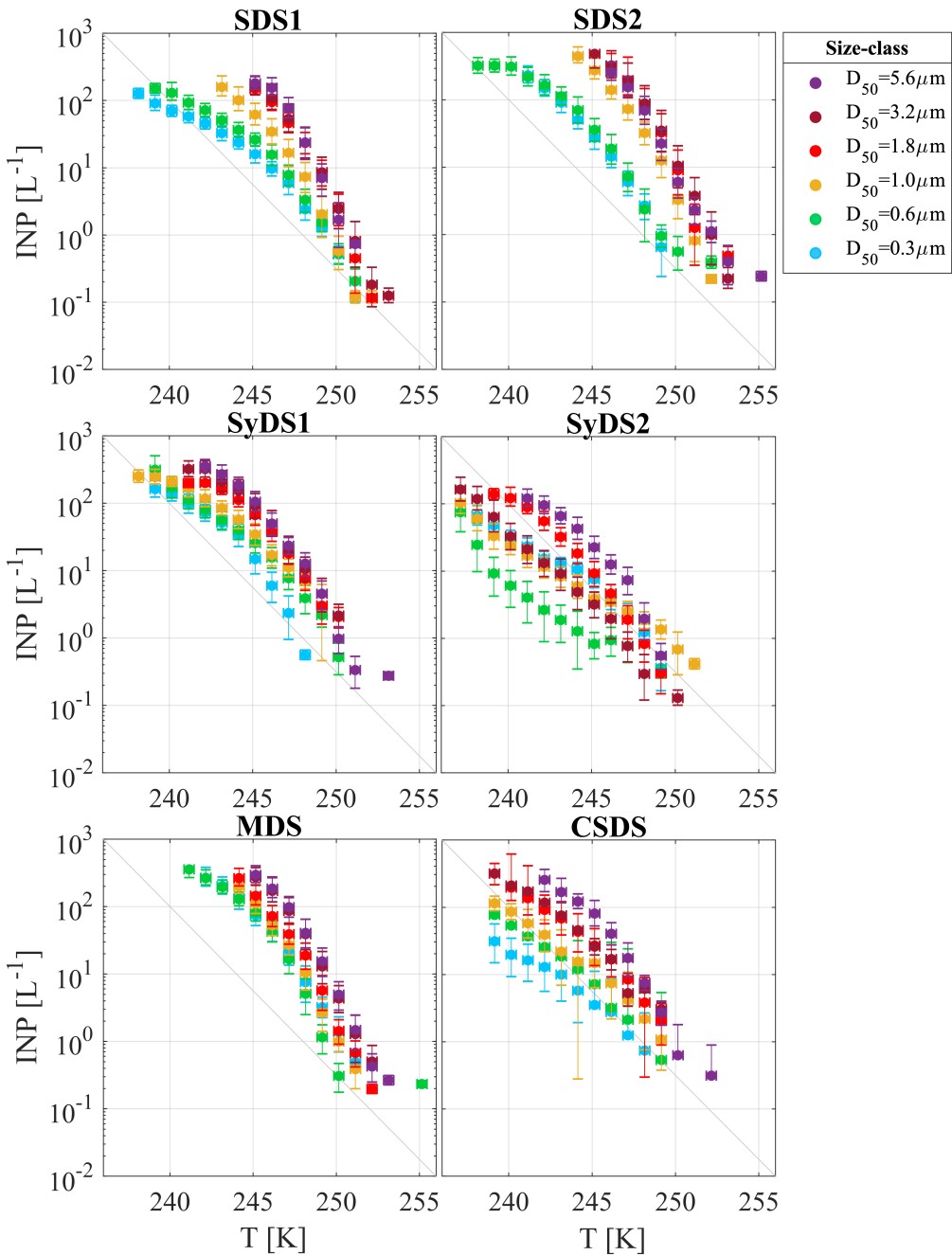


**Figure 4: Airborne INP concentrations measured during dust events. INP concentrations per L$^{-1}$ air as function of temperature, presented in different colors for the different particle size-classes. Uncertainty in temperature is 0.3 K. The grey diagonal line is presented for orientation only.**



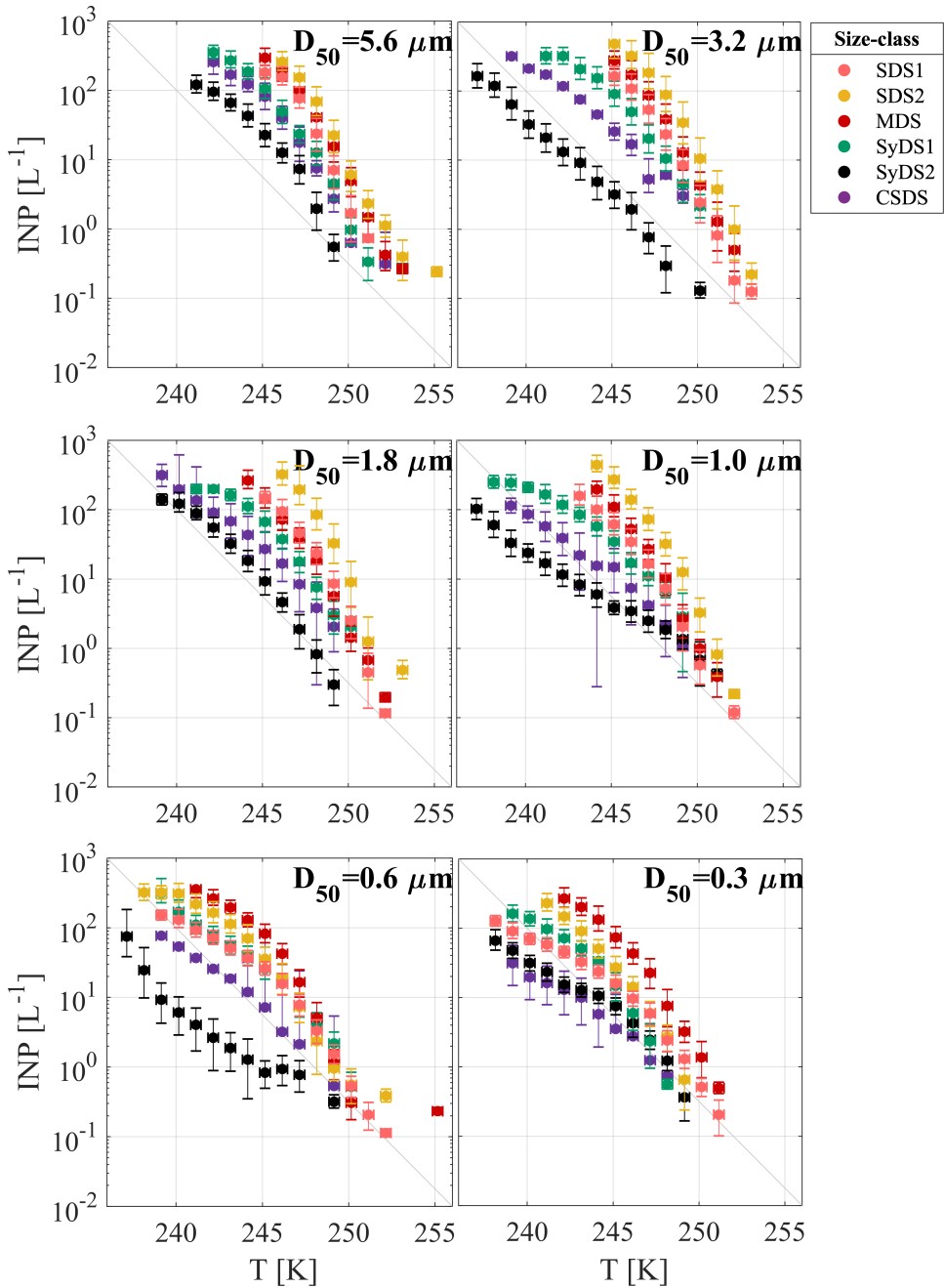


**Figure 5: Airborne INP concentrations for various size classes. INP concentrations per L$^{-1}$ air as function of temperature, presented in different colors for the different dust events that were sampled. Uncertainty in temperature is 0.3 K. The grey diagonal line is presented for orientation only.**

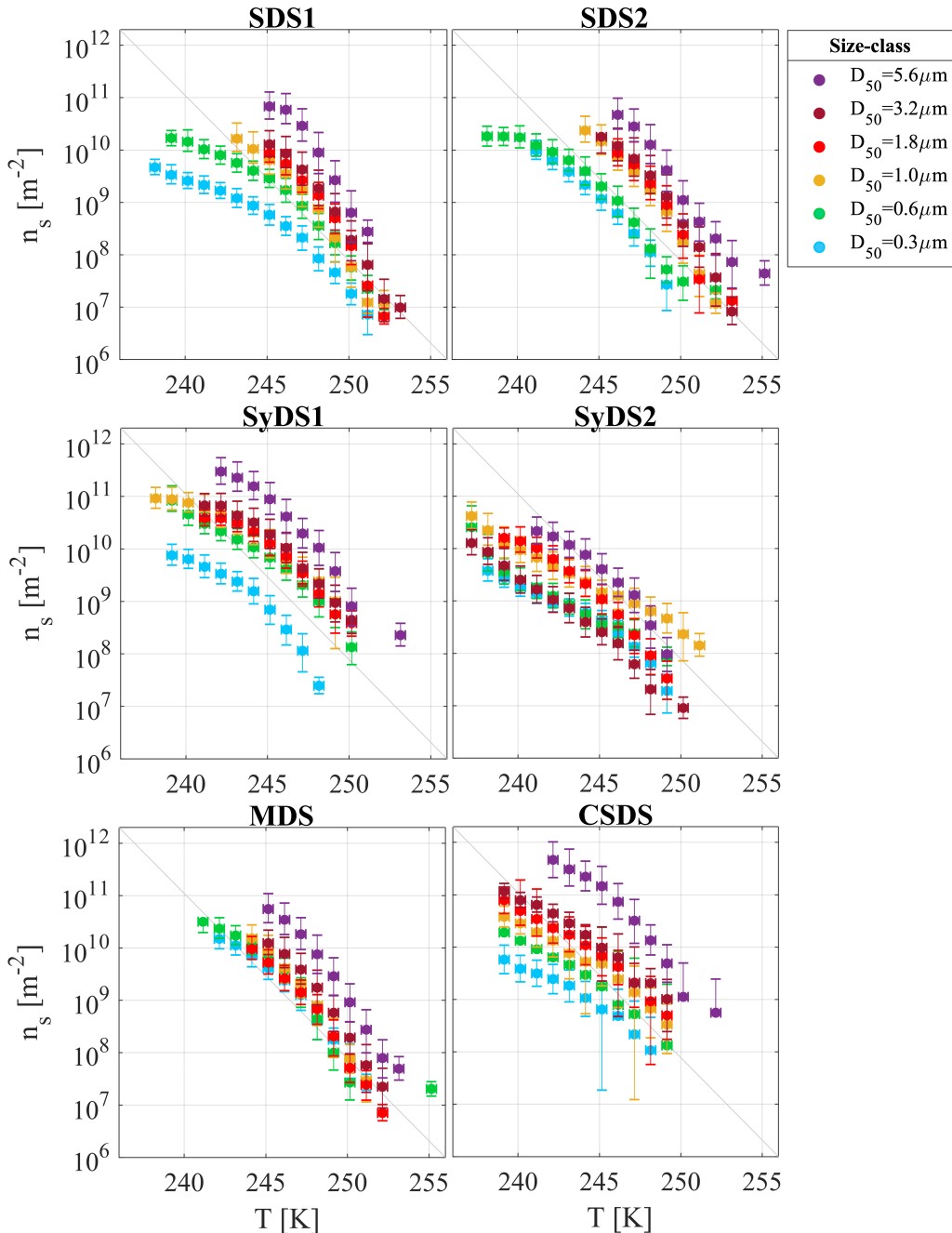

Figure 6: Ice active site density as a function of temperature, $n_s(T)$, for airborne particles dominated by mineral dust are presented individually for each dust event. The different colors represent the different size-classes that were investigated. SDS, SyDS and MDS represent Saharan, Syrian, and mixed dust events, respectively (see text for more details). The linear grey line is identical in each panel to facilitate comparison.

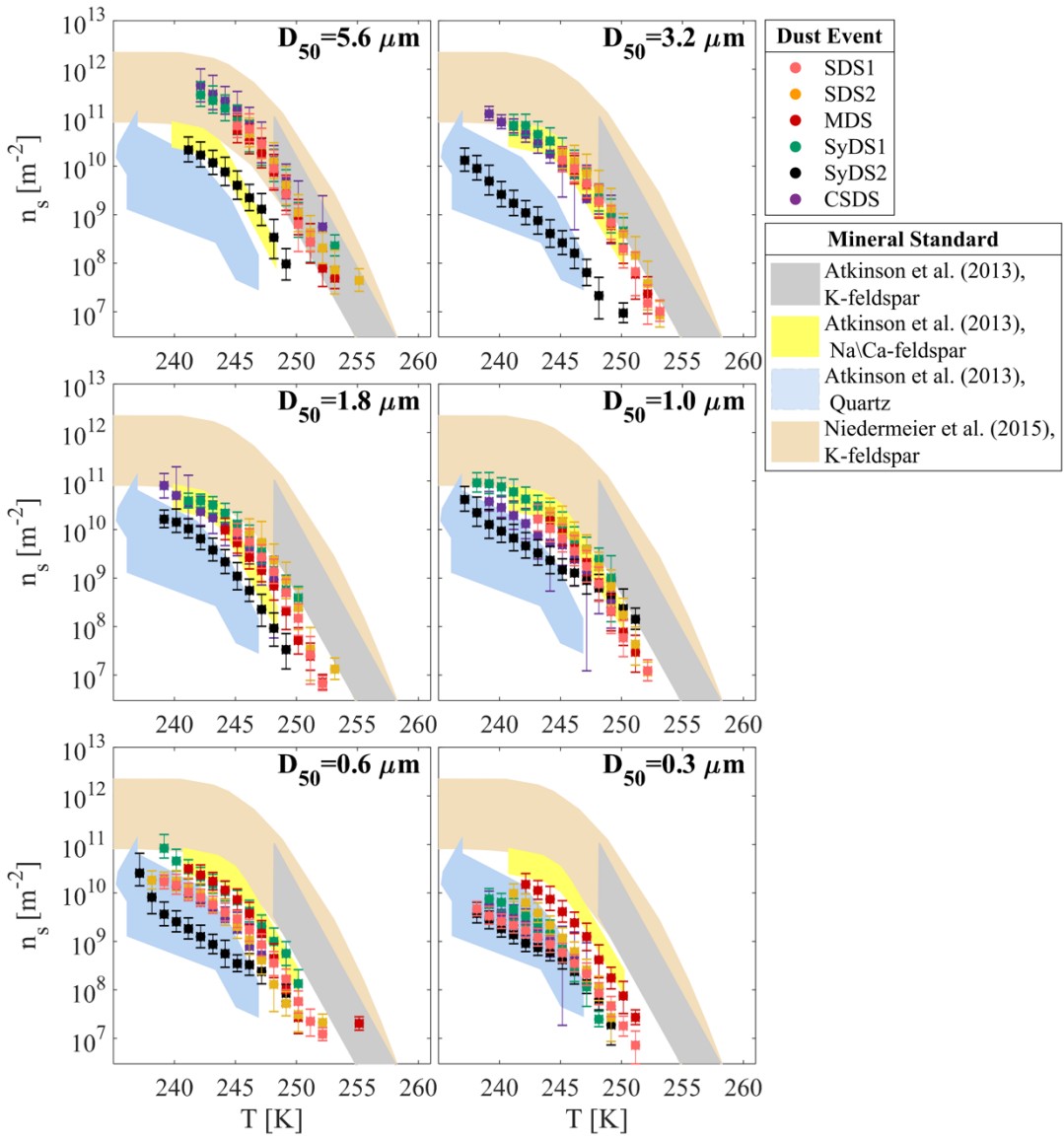

**Figure 7: Ice active site density during dust events in different particle size classes. Dust events from the Sahara Desert (SDS), Syrian Desert (SyDS), or both (MDS) are marked by the different colors. Data for $D_{50}$ = 3.2, 1.8 and 1.0 µm of SDS#2 adopted from Reicher et al. (2018). Relevant standard minerals scaled to ambient values are shown: K-feldspar, Na\Ca-feldspar, and quartz from Atkinson et al. (2013), and K-feldspar from Niedermeier et al. (2015).**



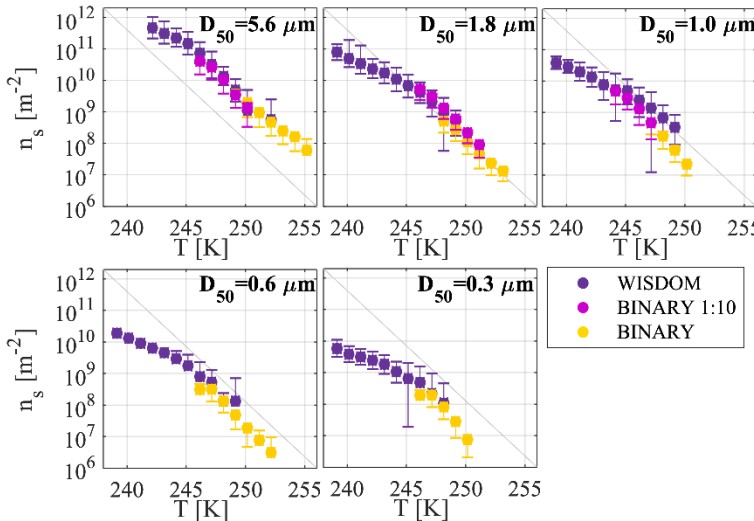


**Figure 8: Complementary measurements of WISDOM and BINARY for CSDS. Analysis in the BINARY was performed to increased detection sensitivity of ice active site densities. Two suspension with different dilution factors were analysed by BINARY and are compared here to the WISDOM data for the different size-classes.**








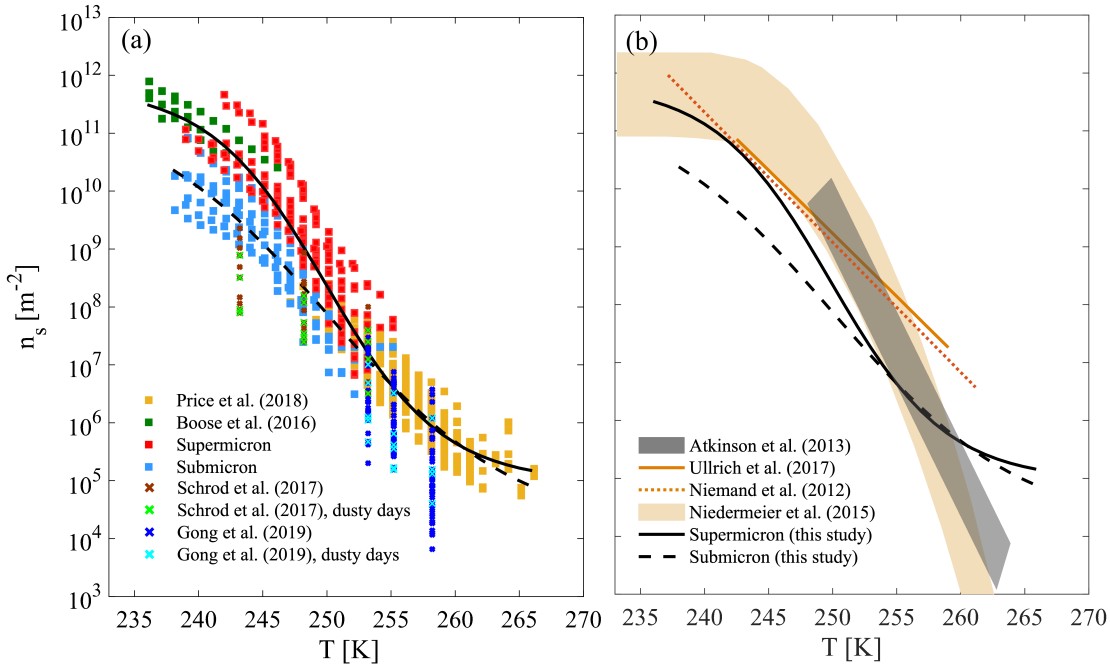

**Figure 9: Heterogeneous ice nucleation by airborne particles during dusty conditions. (a) Active site densities of supermicron and**
**submicron size-classes from this study are shown together with flight data (Schrod et al. (2017) and Price et al. (2018)) and deposited**
**or *in-situ* data (Boose et al. (2016b) and Gong et al. (2019)). New parameterizations, which were derived in this work based on the**
**combined AMD data of the different studies, are shown for supermicron and submicron classes. (b) The new parameterizations**
**derived in this study based on all AMD data, shown next to recent parameterizations for desert dust (Ullrich et al. (2017) and**
**Niemand et al. (2012)) and K-feldspar predictions (Atkinson et al. (2013) and Niedermeier et al. (2015)).**





