# Peer review of "Size-dependent ice nucleation by airborne particles during dust events in the Eastern Mediterranean"

_Atmospheric Chemistry and Physics, 2019_

## Referee Comment (RC1) · Anonymous Referee #1 · 13 Jul 2019

**Review of "Size-dependent ice nucleation by airborne particles during dust events in the Eastern Mediterranean" by Reicher et al.**

**General comment**

This study investigated the ice-nucleating abilities of size-segregated mineral dust particles from seven different events in the Eastern Mediterranean. The ice-nucleating abilities of the collected samples were tested using two different techniques the Welzmann Supercooled Droplets Observation on Microarray (WISDOM) and Bielefeld Ice Nucleation ARraY (BINARY). The authors found a large variability in the INP concentration as a function of the aerosol source. Additionally, supermicron particles were found to be an important contributor to the INP concentration. The authors report that the current data is in agreement with literature data and provided new parametrizations as a function of particle size.

This is an interesting and sound manuscript. The present results are important for the ice nucleation community and can be useful for modelers. The experiments were well designed and were properly executed. The paper nicely fits with the ACP scope and it can be accepted for its publication after the following points are properly addressed.

**Major Comment**

1. The introduction is too short and important information is missing. For example, it is missing what are the characteristics of mineral dust that makes them good INPs. Also, based on the vast literature data, how aging can modify the ice nucleating abilities of mineral dust particles. How much dust is transported to the Mediterranean annually, and when does it happen. It may be important to briefly mention that the presence of mineral dust not only affect cloud formation. What other impacts can be assigned to the presence of mineral dust?
2. I found the inter-comparison of the current data with literature data very selective. I suggest to add other previous studies to provide more robust and solid conclusions.
3. Given that particle size is the focus of this manuscript, I am surprised this was not deeply discussed in the manuscript. What was found before this study in terms of particle size vs. ice nucleating abilities of different aerosol particles? Is the current data in agreement with previous studies? What fraction of the INPs measured in the current study correspond to super-micron size particles?
4. I suggest to improve the English in the revised version.

**Minor comments**

1. Lines 132-133: What is the reason of using two different cooling rates, and why the transition was made at 263K?

2. Line 224 and along the text + Figures: I suggest change the units to L$^{-1}$ as this is commonly used in the ice nucleation community.
3. Lines 246-247: "indicates that the supermicron particles are better INP than the submicron ones". Is this in agreement with literature? Please add a proper discussion here.
4. Lines 238-239: I am wondering if the agreement will be improved if the parametrization is based on total particles and not divided by size. This may be a better comparison with literature data.
5. What is the effect of marine aerosol. The back trajectories show that there is chance that marine particles can affect the ice nucleating abilities of the Saharan dust particles. This is not mentioned/discussed at all.
6. Line 348: I suggest to add a little discussion here on how large or small is this number in comparison to the INP concentrations reported in other environments or parts of the planet (e.g., Polar regions, marine, agricultural, tropics, etc).
7. Figure 9: what of the samples tested in Boose et al. (2016b) are shown here? Pleas add this to the main text as well.

**Technical comments**

Line 34: Add a reference after "troposphere" and after "climate".
Line 36: "Key properties" such as?
Line 37: …in THE characterization.
Line 40: "most prominent". What does it mean?
Line 45-46: "Field observations have identified an increase in INP concentrations and ice clouds formation in the presence of mineral dust". Are these the only 2 studies that found this?
Line 51: Add a reference after "calcite".
Line 55: "and suggested its importance for atmospheric ice formation". This reads a bit awkward.
Line 56: "quartz mineral phases". What does it mean?
Line 61-62: "While there are only few measurements of AMD near source regions". Just Price et al. (2018)?
Line 63: Please add more studies together with the Niemand et al., (2012) study.
Line 64: Add a reference after "AMD".
Line 64: Surface-sampled natural...
Line 65: "to laboratory processes" such as?
Line 72-77: This belongs to the methodology section.
Line 88: I think it is redundant to write MOUDI impactor. I suggest to briefly describe the MOUDI principle.
Line 107: What is close?
Line 112: "particle surface area assumed sphericity and diameter was taken as the midpoint of the GRIMM's channels". This is unclear.
Line 120: "optical diameter > 17.5 µm are assumed to be collected". Is it a good assumption?
Line 122: "of 0.5 µm". Optical diameter?

Line 123: "The initial particle concentration that was used". What does it mean?

Line 126: "Immersion freezing properties". What does it mean?

Line 128: "30 sec" should be 30 s.

Lines 132-133: "10 K min-1" should be 10 K min$^{-1}$.

Line 145: "ice nuclei" should be INP.

Line 145: Do the authors refer to water when talking about a solvent?

Line 157: "ice nucleating particle" should be INP.

Line 170: "travelled directly to the sampling site" from where?

Line 212: "burnings" remove the "s".

Line 217: initiated?

Line 218: Add a reference after "concentrations".

Line 239: "increased with the particle size" delete "the".

Line 240: "the activity was similar" between what?

Line 247: "implying they have better active sites". Better or more? What do the authors mean with better?

Line 256: "important ice-inducing component". What does it mean?

Line 262-264: I found this quite speculative.

Line 324: "Boose et al. (2016)" should be Boose et al. (2016b).

Lines 330 and 333: "$(R - square)$". Fix this.

Line 364: Add a reference after "distance".

Line 365: Add a reference after "scale".

References: DOIs are missing and either use the full name or abbreviated name of the journals. Need to be consistent.

Figure 3: "distributions averaged over the entire sampling periods" events?

Figure 4 and 5: Change the units of the INP concentration to L$^{-1}$.

---

## Referee Comment (RC2) · Anonymous Referee #2 · 16 Jul 2019

General statement

This paper presents results of an experimental investigation of the ice-nucleating properties of aerosol particles sampled from the atmosphere over Israel. Five episodes with mineral dust being transported from the deserts of Northern Africa and the Middle East and one case with clear sky were sampled by MOUDI in six size-classes. Aqueous extracts of the samples were analysed by the droplet freezing devices WISDOM and BINARY.

Atmospheric mineral dust (AMD) is next to sea salt the largest constituent of atmospheric aerosol, and a major ice nucleating agent. Several previous studies have con-

ducted size-resolved INP measurements. It is evident since long ago (e.g. Georgii and Kleinjung, Jour. des Recherches Atmosphériques, 145-156, 1967) that ice nucleating particles (INP) are mostly large particles. This is also found here, and no surprise. However, the new size-resolved data allow a much more detailed understanding of ice nucleation by AMD. Supermicron particles are shown to nucleate at warmer temperatures and to contain higher numbers of INP than submicron particles, even if normalized to the aerosol surface (expressed as surface densities ns of INP). The authors conclude from their ns(T) curves of the different events and size classes, as well as from the overlap with published ns data for minerals, that feldspars dominate the freezing induced by supermicron particles. Quartz dominates ice nucleation by submicron particles. From the comparison with published data it is further concluded that current parameterizations of ns(T) overestimate the activity of airborne dust. As a consequence, the authors derive a new, size-dependent parameterization from their data.

The present manuscript is not just another study on ice nucleation by mineral dust. Its size-resolving approach yields substantial and valuable new information. Including the particle size and the modification of mineral dust during transport in the parameterization – such as done here – will help to improve the modelling of cloud glaciation and related effects.

The work as a whole is sound and perfectly suited to the scope of the journal. The advanced experperimental methods are well documented. The data are well presented and convincingly interpreted in the light of current knowledge and literature. The manuscript as a whole is crafted very well. I recommend publication after some minor adjustments described below.

Major comments

Chapters 2.7.1 and 2.7.2 / sonication times: WISDOM sonicates 90 seconds, whereas BINARY does 30 minutes, accompanied by a 20°C warming. I presume the good

agreement of both methods suggests that the effect of this different treatment is negligible ?

There are some laboratory studies on ice nucleation of minerals that might be cited. Consider to mention and discuss these where relevant, either in the introduction or among the results: 1) Welti et al., Ice nucleation properties of K-feldspar polymorphs and plagioclase Feldspars, Atmos. Chem. Phys. Discuss., https://doi.org/10.5194/acp-2018-1271, 2019

2) Archuleta et al., Ice nucleation by surrogates for atmospheric mineral dust and mineral dust/sulfate particles at cirrus temperatures, Atmos. Chem. Phys., 5, 2617–2634, 2005

3) Lüönd et al., Experimental study on the ice nucleation ability of size-selected kaolinite particles in the immersion mode, J. Geophys. Res., 115, D14201, doi:10.1029/2009JD012959, 2010

Line 161: please spend a few words on how A was derived from primary data.

Line 229: I cannot see from Fig.4 that "SyDS2 has a weaker size dependence in comparison to the other dust events . . ." (smaller spread of curves for a given T), as you state in line 229.

Figures 6 and 7: The clear case CSDS has the highest ns of all data. How is this interpreted? Is the aged tropospheric background aerosol more active than "fresh" mineral dust plumes? Wouldn't that be an interesting result that needs discussion?

Minor comments:

Line 183: maybe add (MDS) after "Another event" ?

Line 184: I believe "west" or "southwest" is better than "south"

Line 211: although it is without consequences, the reader will be interested to know whether the fires are around Rehovot or farther away. Can you say a word on this?

Line 219: maybe add "to ice nucleation" after "supermicron particles"?

Line 234: You write: ".. ranged from 10-3 to almost 1 ..."; I read exactly 10-1 as upper bound.

Line 322: I believe it is "ice cloud formation" or "formation of ice clouds" , instead of "ice clouds formation"

Line 368: Isn't it "emphasizes", because it is related to "overprediction" (singular)?

Technical items

Line 217: Typo: "Ice nucleation is initiated ..." instead of "initiates"

Figure 6, CSDS: The diagonal line is missing in the graph.

Figure 9a) delete "r" in the graph's legend, now it says "(subrmicron class)"

Figure 9: add a) and b) to the left and right graphs

―――――――――――――――――

---

## Author Comment (AC1) · 8 Aug 2019

*The authors are grateful to both of the referees for their positive feeback on the manuscript, and for their valuable and constructive corrections and suggestions that have improved the manuscript. Below we address each of the comments listed in red font. Our answer listed in black font and revised text is listed in green font. Number of lines in our answers are based on the revised manuscript.*

Referee 1:

Review of "Size-dependent ice nucleation by airborne particles during dust events in the Eastern Mediterranean" by Reicher et al. General comment This study investigated the ice-nucleating abilities of size-segregated mineral dust particles from seven different events in the Eastern Mediterranean. The ice-nucleating abilities of the collected samples were tested using two different techniques the WeIzmann Supercooled Droplets Observation on Microarray (WISDOM) and Bielefeld Ice Nucleation ARraY (BINARY). The authors found a large variability in the INP concentration as a function of the aerosol source. Additionally, supermicron particles were found to be an important contributor to the INP concentration. The authors report that the current data is in agreement with literature data and provided new parametrizations as a function of particle size. This is an interesting and sound manuscript. The present results are important for the ice nucleation community and can be useful for modelers. The experiments were well designed and were properly executed. The paper nicely fits with the ACP scope and it can be accepted for its publication after the following points are properly addressed.

Major Comment

1. The introduction is too short and important information is missing. For example, it is missing what are the characteristics of mineral dust that makes them good INPs. Also, based on the vast literature data, how aging can modify the ice nucleating abilities of mineral dust particles. How much dust is transported to the Mediterranean annually, and when does it happen. It may be important to briefly mention that the presence of mineral dust not only affect cloud formation. What other impacts can be assigned to the presence of mineral dust?

The introduction was revised and extended, and now includes more details and references to previous studies.

2. I found the inter-comparison of the current data with literature data very selective. I suggest to add other previous studies to provide more robust and solid conclusions.

We agree that adding more literature data would probably increase the robustness of our conclusions. There are many studies that provide INP concentrations, but $n_s$ values for mineral-dust dominated aerosols in the immersion freezing mode are lacking. Figure 9(a) now includes two studies which measured airborne particles for relatively long time period in the Eastern Mediterranean. Within the sampling period, few Saharan dust events occurred. To Figure 9(b) we added parameterization by Niemand et al (2012) for desert dust, and by Niedermeier et al (2015) for K-feldspar particles.

3. Given that particle size is the focus of this manuscript, I am surprised this was not deeply discussed in the manuscript. What was found before this study in terms of particle size vs. ice nucleating abilities of different aerosol particles? Is the current data in agreement with previous studies? What fraction of the INPs measured in the current study correspond to super-micron size particles?

We now added previous findings to our discussion in section 3.3 (line#251):

"Previous studies also pointed out the significant contribution of supermicron particles to the INP population. Mason et al. (2016) studied the immersion freezing abilities of airborne particles in North America and Europe, and found that supermicron particles dominated the freezing, especially at relatively higher temperature (258 K). Recent measurements in a coastal tropical site conducted by Ladino et al. (2019) also found high concentrations of INPs at relatively high temperatures (> 258 K) due to supermicron particles. In these studies, however, mineral dust is not expected to dominate the samples, and bioaerosol particles are thought to dominate the freezing at the higher temperatures (> 258 K). At lower temperatures (below 253 K), Ladino et al. (2019) suggested that mineral dust dominated the freezing. Moreover, DeMott et al. (2010) found that INP concentrations are correlated with particles > 0.5 μm. Other studies, such as Rosinski et al. (1986) and Huffman et al. (2013), also found that supermicron particles were responsible for most of the INP population in some cases, while when changing the freezing mode that was analysed or the measurement meteorological conditions, their contribution was reduced. Vali (1966) in contrast, found that submicron particles dominate freezing in hail melt samples."

We also updated the conclusions following these additional information (line#416):

"In general, supermicron particles contributed the most to the INP concentration, in agreement with other previous studies (Mason et al., 2016;Huffman et al., 2013;Ladino et al., 2019). However, our current study is probably the only case where mineral dust dominated the samples. Nevertheless, all of these studies highlight the importance of the supermicron size class of AMD for atmospheric ice nucleation."

*4*. I suggest to improve the English in the revised version.
The manuscript has been thorougly edited and correted. We believe that this has improved the readability.

Minor comments
1. Lines 132-133: What is the reason of using two different cooling rates, and why the transition was made at 263K?

The advantage of using two different cooling rates is that the experiment will run faster. Usually, down to 263 K, no freezing activity was observed anyway, and, therefore, there was no need for a high temperature resolution of 1 K per minute. Hence, cooling to 263 K was achieved in 3.5 minutes instead of cooling for 35 minutes (from room temperature ~ 298 K).
The text was revised to explain that (line#159):

"The device was first cooled at a faster constant rate of 10 K min$^{-1}$ from room temperature to 263 K, since freezing events were not expected and indeed were never observed in that temperature range. Then a constant cooling rate of 1 K min$^{-1}$ was used until all the droplets froze."

2. Line 224 and along the text + Figures: I suggest change the units to L-1 as this is commonly used in the ice nucleation community.

Changed.

3. Lines 246-247: "indicates that the supermicron particles are better INP than the submicron ones". Is this in agreement with literature? Please add a proper discussion here.

This is now detailed in the text as shown in point #3 above.

4. Lines 238-239: I am wondering if the agreement will be improved if the parametrization is based on total particles and not divided by size. This may be a better comparison with literature data

As seen in Figure 6, the $n_s$ values of the larger particles ($D_{50}$=5.6) are normaly larger than those of the other particle classes, and – since the graph is on a log scale – summing up would not change the $n_s$ values much.

5. What is the effect of marine aerosol. The back trajectories show that there is chance that marine particles can affect the ice nucleating abilities of the Saharan dust particles. This is not mentioned/discussed at all.

In lines #302-306 we speculated that the passage of dust over the sea could affect its activity and explain why in some cases submicron particles are less efficient than supermicron and in some cases similar. However, we have no way to prove that there was interaction with the marine aerosol and other factors, such as how long it remained over the sea. Information about possible interactions and the effects of the Medditerranean sea is now added to the introduction (line#72):

"Levin et al. (1996) found that AMD particles transported over the Mediterranean Sea were often coated with sulphate and other soluble materials, which could affect clouds' microphysical properties and can eventually result in enhanced ice nucleation."

6. Line 348: I suggest to add a little discussion here on how large or small is this number in comparison to the INP concentrations reported in other environments or parts of the planet (e.g., Polar regions, marine, agricultural, tropics, etc).

Since INP comcentration is not normalised quantity and can be affected by technical issues, such as the amount of material that was immersed in the analysed droplets or the freezing mode that was used, in the revised text we compare our ice-active site density (n$_s$) data, which is more robust, to recent studies, as detailed above in point #2 in major comments.

7. Figure 9: what of the samples tested in Boose et al. (2016b) are shown here? Pleas add this to the main text as well.

We show the four samples of airborne particles. It is detailed in the text in lines #356-357: "Boose et al. (2016b) analysed airborne particles which were deposited in the Eastern Mediterranean region in Egypt, Cyprus and the Peloponnese (Greece) during dust events. Boose et al. (2016b) also sampled airborne particles during dust events over Tenerife, off West Africa."

Technical comments
Line 34: Add a reference after "troposphere" and after "climate".

Added
.
Line 36: "Key properties" such as?

In the revised version we have added details on the "key" properties in the introduction line#52. We did not want to repeat this information, so at that point of the text we added the clarification word 'surface' before "key properties".

Line 37: …in THE characterization.

Added.

Line 40: "most prominent". What does it mean?

We meant to say that it is very dominant and common in the Earth atmosphere. We have reworded the text to "abundant" (line#41):

"One of the most abundant INP in the atmosphere is mineral dust.."

Line 45-46: "Field observations have identified an increase in INP concentrations and ice clouds formation in the presence of mineral dust". Are these the only 2 studies that found this?

Additional appropriate references were added (line#43).

Line 51: Add a reference after "calcite".

Added (line#57).

Line 55: "and suggested its importance for atmospheric ice formation". This reads a bit awkward.

We rephrased the sentence (line#61-64):

"Traditionally, clay minerals were thought to be responsible for atmospheric ice nucleation because they compose much of the dust fraction. However, using standard mineral particles, Atkinson et al. (2013) showed that K-feldspar is the most efficient type, and suggested that it could dominate atmospheric ice formation at relatively high temperatures, above 258 K."

Line 56: "quartz mineral phases". What does it mean?

It means different types of the quartz mineral, but to avoid confusion, we deleted the word "phases".

Line 61-62: "While there are only few measurements of AMD near source regions". Just Price et al. (2018)?

Added (line#80)

Line 63: Please add more studies together with the Niemand et al., (2012) study.

Added (line#82)

Line 64: Add a reference after "AMD".

Added (line#83).

Line 64: Surface-sampled natural...

Rephrased (line#81-82):

"..freezing properties of natural dust or soil samples collected from deserts or standard dust particles."

Line 65: "to laboratory processes" such as?

Rephrased (line#85-86):

"possibly due to laboratory processes, such as milling or sieving, that were applied to the natural dust samples and may have enhanced its activity"

Line 72-77: This belongs to the methodology section.

The methodology details removed from the introduction and the paragraph was rephrased (line#97-101):

"The ability of the collected particles to initiate immersion freezing was studied using the Weizmann Supercooled Droplets Observation on a Microarray (WISDOM) instrument (Reicher et al., 2018), and one of the dust events was studied using the Bielefeld Ice Nucleation ARraY (BINARY) instrument (Budke and Koop, 2015). We characterized the concentrations and the density of ice nucleation active sites (INAS) of AMD in different size-classes for several different dust cases, as well as combined recent literature and available AMD

data to understand how well AMD is represented in models based on recent parameterizations. "

Line 88: I think it is redundant to write MOUDI impactor. I suggest to briefly describe the MOUDI principle.

A short description of the MOUDI impactor is now added to the text (line#112-116): "The MOUDI is a 10-stage impactor with 18 μm cut-point inlet stage followed by size segregating stages with cut points ($D_{50}$) between 0.056 and 10 μm in aerodynamic diameter (Marple et al., 1991). The particles are collected on the different stages as function of their aerodynamic diameter. The collection efficiency for each particle size is described in Marple et al. (1991). Sampling time ranged between 17 and 48 h with a 30 L min$^{-1}$ sample flow rate, similarly to previous studies (Huffman et al., 2013;Mason et al., 2015)."

Line 107: What is close?

The distance is now specified in the text (line#131-133):

"Concentrations of particles with aerodynamic diameters smaller than 10 μm ($PM_{10}$) were measured in the Rehovot station, located about 1 km from our sampling site."

Line 112: "particle surface area assumed sphericity and diameter was taken as the midpoint of the GRIMM's channels". This is unclear.

The text was revised to allow a better description of the method (line#136-138):

"In order to estimate the total surface area that was collected on the different stages, we assumed that the particles are spheres and used the diameter of the GRIMM midpoint of the GRIMM's channels as the particle's diameter."

Line 120: "optical diameter > 17.5 μm are assumed to be collected". Is it a good assumption?

Yes it is a good assumption that is based on the collection efficiency of the MOUDI.

Line 122: "of 0.5 μm". Optical diameter?

Yes. Now specified in the text as well (line#147-148):

…" For example, a small fraction of particles with 0.5 μm optical diameter are collected on stage"..

Line 123: "The initial particle concentration that was used". What does it mean?

Since we analyzed a filter that collected particles for a few hours, we summed up all the OPC readings to obtain the total number size distribution of the particles that were collected, and this is the initial particle concentration.

Line 126: "Immersion freezing properties". What does it mean?

Rephrased to "Immersion freezing abilities of the sampled ambient particles" (line#152).

Line 128: "30 sec" should be 30 s.

Corrected.

Lines 132-133: "10 K min-1" should be 10 K min-1.

Corrected.

Line 145: "ice nuclei" should be INP.

Corrected.

Line 145: Do the authors refer to water when talking about a solvent?

Yes, in othis study the solvent it is water.  It is detailed in Eq. #1.

Line 157: "ice nucleating particle" should be INP.

Corrected.

Line 170: "travelled directly to the sampling site" from where?

We meant to emphasize that in some cases there was a short and direct path to the sampling site, while in others, the path was longer, for example, where the air mass was deflected to the sampling site while travelled in a different path. We rephrased the sentence (line#199):

"...the air masses travelled directly to the sampling site from the source region."

Line 212: "burnings" remove the "s".

Corrected.

Line 217: initiated?

Corrected.

Line 218: Add a reference after "concentrations"

This is a claim we made.
Line 239: "increased with the particle size" delete "the".

Corrected.
Line 240: "the activity was similar" between what?

The activity of the dust in the three different stages is similar, considering the measurement uncertainties. The sentence was rephrased (line#279-280):

"The highest $n_s$ values were observed in the supermicron range $D_{50}$=5.6 µm, followed by $D_{50}$=3.2, 1.8 and 1.0 µm. The activity of the latter three classes was similar within measurement uncertainties."

Line 247: "implying they have better active sites". Better or more? What do the authors mean with better?

'better' is now explained in the text (line#287-288):
"they have more active sites or/and active sites that nucleate ice at higher temperatures.".

Line 256: "important ice-inducing component". What does it mean?

It means that from a mixture of minerals that the particles are composed of, there is one componenet that dominates the observed ice nucleation (i.e., these particles are the most active ones).

Line 262-264: I found this quite speculative.

We updated the text so that it will be clear that this is our speculation (line#302-306):

"For example, we propose that the passage of SDS1 and SDS2 over the Mediterranean Sea can contribute to their reduced activity in the submicron range, while for the MDS event, a shorter and relatively direct transport path resulted in less atmospheric processing. Although speculative, these considerations may possibly explain why the freezing activity of submicron particles converged with those of the supermicron particles, but we acknowledge that further measurements are needed to confirm these suggestions."

Line 324: "Boose et al. (2016)" should be Boose et al. (2016b).
Corrected.
Lines 330 and 333: "$(R - square)$". Fix this.

Corrected.

Line 364: Add a reference after "distance".

Added (line#421).

Line 365: Add a reference after "scale".

Added (line#423).

References: DOIs are missing and either use the full name or abbreviated name of the journals. Need to be consistent.

Corrected.

Figure 3: "distributions averaged over the entire sampling periods" events?

For each event, the average size distribution is shown. Clarified now in the text: "..distributions averaged over the entire sampling periods of the events as monitored by GRIMM OPC during the studied events."

Figure 4 and 5: Change the units of the INP concentration to L-1 .

 Changed.

Referee 2:

**General statement**

This paper presents results of an experimental investigation of the ice-nucleating properties of aerosol particles sampled from the atmosphere over Israel. Five episodes with mineral dust being transported from the deserts of Northern Africa and the Middle East and one case with clear sky were sampled by MOUDI in six size-classes. Aqueous extracts of the samples were analysed by the droplet freezing devices WISDOM and BINARY. Atmospheric mineral dust (AMD) is next to sea salt the largest constituent of atmospheric aerosol, and a major ice nucleating agent. Several previous studies have conducted size-resolved INP measurements. It is evident since long ago (e.g. Georgii and Kleinjung, Jour. des Recherches Atmosphériques, 145-156, 1967) that ice nucleating particles (INP) are mostly large particles. This is also found here, and no surprise. However, the new size-resolved data allow a much more detailed understanding of ice nucleation by AMD. Supermicron particles are shown to nucleate at warmer temperatures and to contain higher numbers of INP than submicron particles, even if normalized to the aerosol surface (expressed as surface densities ns of INP). The authors conclude from their ns(T) curves of the different events and size classes, as well as from the overlap with published ns data for minerals, that feldspars dominate the freezing induced by supermicron particles. Quartz dominates ice nucleation by submicron particles. From the comparison with published data it is further concluded that current parameterizations of ns(T) overestimate the activity of airborne dust. As a consequence, the authors derive a new, size-dependent parameterization from their data. The present manuscript is not just another study on ice nucleation by mineral dust. Its size-resolving approach yields substantial and valuable new information. Including the particle size and the modification of mineral dust during transport in the parameterization – such as done here – will help to improve the modelling of cloud glaciation and related effects. The work as a whole is sound and perfectly suited to the scope of the journal. The advanced experimental methods are well documented. The data are well presented and convincingly interpreted in the light of current knowledge and literature. The manuscript as a whole is crafted very well. I recommend publication after some minor adjustments described below.

Major comments

Chapters 2.7.1 and 2.7.2 / sonication times: WISDOM sonicates 90 seconds, whereas BINARY does 30 minutes, accompanied by a 20°C warming. I presume the good agreement of both methods suggests that the effect of this different treatment is negligible?

Yes, we do not think the different sonication protocols have any significant effect on the ice nucleating properties of the dust. The sonication used for preparing the suspensions analyzed in WISDOM was a dry sonicator and, hence, this is more intense than the sonicator used for suspension preparation for the BINARY experiments, which was a bath sonicator (much of the energy is lost during the sonication to the water bath). This is the reason for the chosen longer sonication time in the BINARY. We have added some additional explanatory text to section 2.7.1 accordingly (line#153-154):

 "Immersion freezing activity of the sampled ambient mineral dust was measured using suspensions of the collected particles that were extracted from the filters by dry sonication (VialTweeter, model UP200St; Hielcher). This type of sonication method is more effective than the ultrasonic bath in which most of the energy is dissipates in the surrounding water."

There are some laboratory studies on ice nucleation of minerals that might be cited. Consider to mention and discuss these where relevant, either in the introduction or among the results: 1) Welti et al., Ice nucleation properties of K-feldspar polymorphs and plagioclase Feldspars, Atmos. Chem. Phys. Discuss., https://doi.org/10.5194/acp2018-1271, 2019 2) Archuleta et al., Ice nucleation by surrogates for atmospheric mineral dust and mineral dust/sulfate particles at cirrus temperatures, Atmos. Chem. Phys., 5, 2617–2634, 2005 3) Lüönd et al., Experimental study on the ice nucleation ability of size-selected kaolinite particles in the immersion mode, J. Geophys. Res., 115, D14201, doi:10.1029/2009JD012959, 2010

Thank you. These references were relevant and have been added to the text (line#51)

Line 161: please spend a few words on how A was derived from primary data.

It is explained in section 2.5 (line#134), and we added a clarification in the text (line#189-190):

"…A is the surface area immersed in a single droplet of the experiment, based on the total surface area of particles in the suspension."

Line 229: I cannot see from Fig.4 that "SyDS2 has a weaker size dependence in comparison to the other dust events . . ." (smaller spread of curves for a given T), as you state in line 229.

For SyDS2, in the supermicron range, particles in the stage of $D_{50}$ =3.2 μm had lower activity (and also different slope) than in the stage of $D_{50}$=1.8 μm, and in the warmer temperatures (above 247 K), also lower activity than particles in the stage of $D_{50}$=1.0 μm. In the submicron range, particles of stage of $D_{50}$= 0.6 μm had lower activity than particles in the stage of $D_{50}$ =0.3 μm. In the other events, the dependency of the ice nucleation activity in the particle size was kept, and normally larger particles were more active than the smaller ones.

Figures 6 and 7: The clear case CSDS has the highest ns of all data. How is this interpreted? Is the aged tropospheric background aerosol more active than "fresh" mineral dust plumes? Wouldn't that be an interesting result that needs discussion?

The CSDS event had less surface area in the suspension than the rest of the SDS events. We believe that this is the reason for the higher $n_s$ values: when there is less active material within the droplets, then the droplets experience higher supercooling because there are fewer active sites at warmer temperatures. Therefore, some of the droplets were allowed to cool to colder temperature. As the temperature becomes colder, higher numbers of nucleation sites were activated, and therefore higher $n_s$ values. In Figure 7 for example, CSDS collapse to the same activity of the other events, and the activity is similar ($D_{50}$ = 5.6,3.2,1.8,0.6,0.3) or smaller than most of the events ($D_{50}$ = 1.0).

Minor comments:

Line 183: maybe add (MDS) after "Another event"?

Yes, it is more logical to introduce the event here, now the text was revised (line#213):

"Another event was defined as a "mixed dust" event (MDS), because it was more complicated and included contributions of different sources."

Line 184: I believe "west" or "southwest" is better than "south"

Corrected to 'southwest' (line#215).

Line 211: although it is without consequences, the reader will be interested to know whether the fires are around Rehovot or farther away. Can you say a word on this?

We added to the text the information about the radius in which the fires occurred (line#241):

"Note that prior to and during this event, a series of biomass burning events occurred in Israel extending to about 100 km north and 50 km east of the sampling site."

Line 219: maybe add "to ice nucleation" after "supermicron particles"?

Added.

Line 234: You write: ".. ranged from 10-3 to almost 1 . . ."; I read exactly 10-1 as upper bound.

Thank you, this is indeed correct, the value is almost 0.3 and this is closer to $10^{-1}$ than to 1. Value was changed in the text from 1 to $10^2$ (on the Liter scale) (Line#263-264).

 Line 322: I believe it is "ice cloud formation" or "formation of ice clouds", instead of "ice clouds formation"

Corrected to "ice cloud formation" (Line#375).

Line 368: Isn't it "emphasizes", because it is related to "overprediction" (singular)?

Yes, thank you, corrected.

Technical items

Line 217: Typo: "Ice nucleation is initiated . . ." instead of "initiates"

Corrected.

Figure 6, CSDS: The diagonal line is missing in the graph.

Fixed.

Figure 9a) delete "r" in the graph's legend, now it says "(subrmicron class)"

Fixed.

Figure 9: add a) and b) to the left and right graphs

Fixed.

[revised manuscript text omitted]

---

## Author Response (AR2)

August 14, 2019

Prof. Allan Bertram – Editor

Dear Allen,

Thank you for handling our manuscript. We have made the corrections you have asked for in the latest version and hope that now the paper can be accepted.

Keep well
Yinon Rudich